# *Urtica dioica* Aqueous Leaf Extract: Chemical Composition and In Vitro Evaluation of Biological Activities

**DOI:** 10.3390/ijms26031220

**Published:** 2025-01-30

**Authors:** Nouha Dakhli, Auxiliadora López-Jiménez, Casimiro Cárdenas, Manel Hraoui, Jihene Dhaouafi, Manuel Bernal, Hichem Sebai, Miguel Ángel Medina

**Affiliations:** 1Laboratory of Functional Physiology and Valorization of Bio-Ressources, Higher Institute of Biotechnology of Beja, University of Jendouba, Beja 382-9000, Tunisia; nouhadakhli88@gmail.com (N.D.); hraouimanel@gmail.com (M.H.); jihen.dhaouafi.etu@univ-lille.fr (J.D.); 2Research Support Central Services (SCAI), University of Málaga, E-29071 Málaga, Spain; alj@uma.es (A.L.-J.); ccg@uma.es (C.C.); 3UMR Transfrontalière BioEcoAgro N°1158, Université Lille, INRAE, Université Liège, UPJV, YNCREA, Université Artois, Université Littoral Côte d’Opale, ICV-Institut Charles Viollette, F-59000 Lille, France; 4Department of Molecular Biology and Biochemistry, Faculty of Sciences, University of Málaga, Andalucía Tech, E-29071 Málaga, Spain; mbernal@uma.es; 5Málaga Biomedical Research Institute and Nanomedicine Platform (IBIMA-Plataforma BIONAND), C/Severo Ochoa, 35, E-29590 Málaga, Spain; 6Network Biomedical Research Center for Rare Diseases (CIBERER), U741, E-28029 Málaga, Spain

**Keywords:** *Urtica dioica*, antibacterial activities, antioxidant activities, antitumoral activities, HCT-116, colon cancer

## Abstract

*Urtica dioica* L. has been used as a natural remedy due to its healing properties for over 2000 years. The aim of this study is to investigate the chemical composition, antimicrobial, antioxidant, and antitumor properties in vitro of the aqueous extract of *Urtica dioica* leaves (AEUD). The chemical composition was assessed by an ultra-high-performance liquid chromatography system coupled to a benchtop QExactive high-resolution accurate mass spectrometry operating in a data-dependent acquisition mode as a non-target approach. Minimal inhibitory concentration (MIC) and disc diffusion were used to assess the antibacterial efficacy against nine bacterial strains. The antioxidant impact was assessed using DPPH, ABTS, FRAP, and ferrous ion-chelating ability assays. By using the MTT method, the cytotoxicity effect of AEUD on colon cancer cell HCT-116 was evaluated. Flow cytometry was used to analyze the cell cycle. Finally, the anti-migration and anti-invasion properties of AEUD on HCT-116 cells were estimated using the wound healing test and Transwell assays. AEUD is a rich source of phenolic compounds. The results of disc diffusion and MIC showed that the AEUD is more active against Gram-positive bacteria than against Gram-negative bacteria. MTT assay confirmed that the AEUD inhibited HCT-116 colon cancer cell proliferation. Findings of flow cytometry confirmed that cell cycle arrest occurred at the G2 phase. Additionally, AEUD had anti-migration and anti-invasion effects. This study shows that *Urtica dioica* aqueous leaf extract exhibits potential antibacterial, antioxidant, and antitumoral activities on HCT-116 colon cancer cells.

## 1. Introduction

*Urtica dioica* L. (UD), a perennial herbaceous plant with spiny leaves that is a member of the nettle family (Urticaceae), is known as the stinging nettle or common nettle. For ages, people have used it as a natural remedy due to its healing properties for over 2000 years [1,2]. Stinging nettles are commonest in Europe, North America, North Africa, and some regions of Asia, though they can be found practically anywhere.

Herbal medications are frequently recommended due to their perceived efficiency, little adverse effects in clinical practice, and comparatively inexpensive cost [3], even in the case that their biologically active components are unknown.

Despite the significant advancements in therapeutic research, there is still a demand for strong and efficient analgesic medications. In this regard, it has been widely demonstrated that a variety of plant-based compounds play an important part in the process of developing new treatment methods [4].

Nettle has been considered as a dish or portion of food with medicinal qualities that is meant to both prevent and treat illnesses [5]. Young leaves are a healthy culinary herb that can be used in herbal therapy as well as cooked and eaten [6]. Lignans, secolignans, norlignans, alkaloids, sesquiterpenoids, flavonoids, triterpenoids, sphingolipids, and sterols are among the compounds that have been found in this plant [7]. The trichomes of the nettle are suspected to contain formic acid, acetyl choline, serotonin, and histamine [6]. Nettle is a highly nutritive food that is rich in minerals (particularly iron), vitamin C, and pro-vitamin A [8]. This plant may also influence protein and lipid metabolism and enhance their functionality [9]. Studies have shown that the stinging nettle plant contains physiologically active compounds that can help to decrease the production of free radicals brought on by the conditions of modern lifestyles [6].

Previously claimed health benefits of stinging nettle include anti-proliferative, anti-inflammatory, antioxidant, analgesic, immunological stimulatory, anti-infectious, hypotensive, anti-ulcer, anti-cancer, and cardiovascular disease prevention [10].

The entire plant is used in folk medicine to cure a variety of conditions, including diabetes, kidney stones, burns, anemia, rashes, and internal bleeding [11]. Only a small number of these pharmacological actions, though, have been demonstrated by experimentation [12].

Additionally, stinging nettle is occasionally used in cosmetics and is employed as a source of fiber for textiles [13]. Chlorophyll, an E140 green coloring component used in foods and pharmaceuticals, is extracted from plants for commercial use [14].

New therapeutic approaches may become possible by better understanding the molecular mechanisms causing positive outcomes [15]. This is perhaps the reason why it has recently attracted the attention of scientists once again, leading to an increase in the number of studies that confirm the positive benefits of *Urtica dioica* on people all over the world [16].

In the present work, we conducted a comprehensive phytochemical and biological characterization of aqueous extract of *Urtica dioica* leaves (AEUD). For this purpose, the qualitative biochemical profiling of *Urtica dioica* leaves was determined using an ultra-high-performance liquid chromatography system coupled to a benchtop QExactive high-resolution accurate mass spectrometry operating in a data-dependent acquisition mode as a non-target approach, which is designed for the discovery and characterization of unknown compounds in complex samples. On the other hand, the biological activities were evaluated by different in vitro assays: (i) antioxidant activities were evaluated by DPPH, ABTS, FRAP, and ferrous ion-chelating ability assays; (ii) minimal inhibitory concentration (MIC) and disc diffusion were used to assess the antibacterial efficacy against nine bacterial strains; and (iii) the antitumor activity on colon cancer cells HCT-116 proliferation, migration and invasion were also studied.

## 2. Results

### 2.1. Chemical Profile of the Aqueous Extract of Urtica dioica Leaves

To find out the bioactive compounds responsible for the biological activities in *Urtica dioica* aqueous leaf extract, the total ion chromatogram (TIC) for the acquired sample is given in Figure 1. Compounds identified in *Urtica dioica* aqueous leaf extract are presented in Table 1. Detailed information related to the identification and confirmation of detected compounds is presented in Appendix A.

### 2.2. Antimicrobial Activity

In this study, we screened the antibacterial activity of AEUD against nine bacterial strains, and its activity potentials were qualitatively and quantitatively assessed by the presence or absence of inhibition zones and MIC values.

The disc diameters of the inhibition zone of AEUD against the tested bacteria are shown in Figure 2 and Table 2. The obtained results indicated that the extract displayed a variable degree of antibacterial activity on the different tested strains. The highest activity was observed against *S. typhimirium* ATCC 14028 with the largest zone (14 mm), followed by *P. aeruginosa* ATCC 27853 (13 mm) and *E. faecalis* ATCC 29212 (10 mm). Tested strains were more sensitive to gentamicin (30–70 mm) than to the AEUD tested.

Our results were confirmed by the MIC values (Table 3). The inhibitory properties of the AEUD were observed within a range of concentrations from 0.195 to 12.5 mg/mL. AEUD showed the lowest concentration of MIC value of 0.19 mg/mL against *E. coli* and 0.39 mg/mL, against *P. aeruginosa, E. faecalis,* and *Bacillus cereus.*

Results shown in Table 3 indicated that aqueous extract from *Urtica dioica* has a bactericidal effect against *Salmonella thyphimurium* ATCC 14028, *Salmonella enteridis* ATCC 13076, *Shigella*, *Listeria monocytogenes* ATCC 7644, *Enterococcus faecalis* ATCC 29212 and *Bacillus cereus* ATCC 11778. It has a bacteriostatic effect against *P. aeruginosa* ATCC 27853, *E. coli* ATCC 25983, and *S. aureus* ATCC 29213.

### 2.3. Antioxidant Activity

In the present study, we investigated the antioxidant activity of the aqueous extract of *Urtica dioica*. The DPPH, ABTS, FRAP, and ferrous ion-chelating ability assays have been used to evaluate the free radical scavenging capacity of the AEUD. Effective antioxidant activity was observed for the AEUD in all assays (Figure 3).

### 2.4. Cytotoxic Effects of AEUD on HCT-116 Tumor Cells

To investigate the antitumoral activity of AEUD, we first evaluated its effect on cell proliferation of colon cancer cell line. MTT assay, which is employed for primary screening of new drugs with efficient effects on cell survival and/or proliferation, was used to examine the cytotoxic effects of the AEUD on HCT-116 cancer cells. The survival curves of HCT-116 treated for three days with various dosages of AEUD are shown in Figure 4. In vitro, cytotoxic effects of AEUD against HCT-116 colon cancer cells indicated that the extract had an antiproliferative activity.

The IC_50_ concentration of *Urtica* extract on HCT-116 cells estimated from dose-response survival curves was 2.35 ± 0.77 mg/mL.

### 2.5. AEUD Induces Apoptosis of HCT-116 Colon Cancer Cells

Based on the cytotoxicity results, we examined the cell cycle phases of colon cancer cells after treatment with three concentrations of AEUD (1, 2, and 4 mg/mL) by using suspensions of permeabilized cells stained with propidium iodide, which allows for the easy analysis of the cellular population distribution in the different phases of the cell cycle by flow cytometry. A significant increase in the percentage of cells in the SubG1 phase of treated cells is an indication of a possible pro-apoptotic effect of the treatment. AEUD caused a dose-dependent increase in the percentage of cells in the SubG1 phase (Figure 5).

The proportion of cells at the S/G2/M phase was greater as compared to the control group, indicating that AEUD effectively induced S/G2/M phase cell cycle arrest, which subsequently continued with a decrease in cell population at G0/G1 phases. These results clearly demonstrated that AEUD can significantly stop the cell cycle at the G2 phase and inhibit the progressive growth of the cancer cells (Figure 5B).

### 2.6. AEUD Inhibits Migration of HCT-116 Tumor Cells

An essential sign of malignant cells that may infect surrounding tissues or move through blood arteries is metastasis. We therefore used tests for wound healing to investigate these properties. According to our findings, AEUD inhibits weakly the ability of HCT-116 colon cancer cells to migrate (Figure 6).

### 2.7. AEUD Inhibits Invasion of HCT-116 Tumor Cells

A transwell invasion experiment was used to assess how AEUD affected the capacity of HCT-116 cells to migrate in a chemoattractant gradient. Figure 7A demonstrates a dose-dependent decrease in the invasiveness of tumor cells treated with AEUD in comparison to controls, which has been corroborated by their quantification (Figure 7B).

## 3. Discussion

In recent years, the main aim of many ethnopharmacological research projects has been to find natural bioactive compounds with biological activities in plants traditionally used against diseases. In this study, we evaluated the qualitative profiling of *Urtica dioica* leaves and tested the potential of the aqueous extract of *Urtica dioica* to produce antibacterial, antioxidant, and antitumor effects by using different in vitro assays.

Ultra-high-performance liquid chromatography system coupled to a benchtop QExactive high-resolution accurate mass spectrometry operating in a data-dependent acquisition mode as a non-target approach analysis was used to obtain qualitative profiling of compounds in *Urtica dioica* L. leaf extract. Compounds detected were 2-hydroxycinnamic acid, 3,4-dihydroxybenzaldehyde, 4-hydroxycoumarine, 4-methylumbelliferone hydrate, alpha-bisabolol, alpha-bisabolol acetate, alpha-farnesene, alpha-pinene, angelic acid, azulene, bisabolol oxide a, caffeic acid, catechol, chamazulene, ethyl protocatechuate, and p-coumaric acid. The findings of the current study’s chemical profile of AEUD may aid in understanding the chemical agents and components that underlie its biological impacts.

In a previous research article, the phenolic compounds in *Urtica dioica* L. leaf extract detected were 2-O-caffeoyl malic acid, cholrogenic acid, p-coumaroyl malic acid, caffeic acid, rutin, isoquercetin, kaempferol 3-O-rutinoside, isorhamnetin 3-O-rutinoside, and isorhamnetin hexoside, while the most abundant were2-O-caffeoyl malic acid, chlorogenic acid, and rutin [17]. The amount of rutin in dried *Urtica dioica* L. leaf extract was significantly higher than that in leaf extract made from *Urtica dioica* L. plant material that was gathered for studies in Serbia, according to data from the literature [18]. In addition to tannins, volatile and fatty acids, polysaccharides, isolectins, sterols, terpenes, and proteins, nettle leaves are rich in flavonoids, phenolic compounds, organic acids, vitamins, and minerals [10,14,19,20].

In a different investigation, the extract was discovered to contain patuletin, a recognized O-methylated flavonol [21], and quercetagetin [22]. M/p-hydroxybenzoic acid, a phenol derivative of benzoic acid that is regarded as a significant dietary phenolic agent, was another substance that was prevalent in the extract [23]. Other substances found in trace levels included homovanillic acid, gallic acid, chlorogenic acid, and bisabolol oxide B [24]. The primary bioactive components of AEUD identified by HPLC-DAD analysis are flavonoids and hydroxycinnamic acid derivatives [25,26].

The composition of *U. dioica* in this study, however, is different from that stated by other authors can be explained that planting, climatic, seasonal, and experimental settings can all affect the content of an extract of any plant [27]. Additionally, this can be the result of variations in plant materials, which might change significantly depending on regional growing circumstances and varietal variants [28].

This study evaluated the antimicrobial activity of the aqueous extract of nettle against nine pathogenic strains. Our results showed that the tested microorganisms had relatively low MIC values ranging from 0.195 to 12.5 mg/mL. However, lower MIC was observed against Gram-positive bacteria including *S. aureus, E. faecalis, B. cereus,* and *L. monocytogenes*, while relatively less antibacterial activity was associated with Gram-negative bacteria including *E. coli*, *S. typhimurium*, *S. enteritidis*, *P. aeruginosa,* and *S. flexneri*.

It can be seen that *U. dioica* is active against Gram-positive bacteria more than Gram-negative bacteria. In general, Gram-negative bacteria are more resistant than Gram-positive [29]. In fact, Gram-negative bacteria have an outer membrane envelope that acts as a barrier to the entrance of external agents and their porins determine the type and size of these substances that can reach their cytoplasm [30,31]. Consequently, the permeability of the membrane of these bacteria is much less than that of Gram-positive bacteria [32], as confirmed by many studies [33,34,35]. Similarly, another study showed that an aqueous extract of *U. dioica* had potent antimicrobial activity against different microorganisms, including *E. coli*, *Enterobacter aerogenes*, *S. epidermidis,* and *Candida albicans.*

In addition, the main bacterial effect observed was bactericidal, which is important from a pharmaceutical point of view. Moreover, the obtained results are of great importance, particularly in the case of *S. aureus*, *B. cereus*, *E. coli*, and *S. typhimirium* which are well-known for being resistant to antibiotics and for the production of several types of enterotoxins that cause gastroenteritis infections [36,37,38].

The antibacterial activities of the AEUD could be attributed to the presence of various constituents, including chlorogenic acid, caffeic acid, rosmarinic acid, and rutin.

We also investigated the AEUD’s antioxidant capacity. Numerous extracts have been tested to see how well they can scavenge free radicals using the DPPH, ABTS, FRAP, and ferrous ion-chelating ability assays. Although the basic concepts are similar, the ABTS scavenging assay is preferred because it can assess the antioxidant activity of both lipophilic and hydrophilic antioxidants. The antioxidant activity measurements in DPPH and ABTS assays are quick, sensitive, and most frequently applied for the preliminary assessment of the antioxidant potential of various natural substances. The flexible FRAP test can be used to measure an antioxidant compound’s ability to reduce the ferric tripyridyl triazine (Fe^3+^-TPTZ) complex. All the tested methods proved that the AEUD had a high antioxidative capacity compared to the reference molecule. Radical scavenging and reducing the power ability of the aqueous extract is usually associated with a high content of phenolic components.

Our results are consistent with research from southern Italy that demonstrates the antioxidant efficacy of UD leaf extract. In this study, 27 non-cultivated vegetables, including UD, were examined for their ability to scavenge free radicals using the 1,1-diphenyl-2-picrylhydrazil radical (DPPH) assay, as well as their ability to prevent lipid peroxidation in liposomes and xanthine oxidase [39]. In a recent work, ABTS and DPPH tests were used to examine the antioxidant activity of extract from three plants (*Urtica dioica*, *Matricaria chamomilla,* and *Murraya koenigii*). The UD was shown to have lesser antioxidant activity than *Murraya koenigii* but higher antioxidant activity than *Matricaria chamomilla* in both assays [40]. In another study, antioxidant and antibacterial activity were investigated using an extract of UD leaves. The non-enzymatic antioxidant activity of the extract was investigated using ferric reducing antioxidant power (FRAP), DPPH, and ABTS assays. The leaves of UD showed effective antioxidant activity in all experiments [41]. In an Iranian investigation, an aqueous extract of UD leaves was used to examine the antioxidant activity before analyzing the anticancer effect utilizing the 3-(4,5-dimethylthiazole-2-yl)-2,5-diphenyltetrazoliumbromide (MTT) and FRAP techniques. The scientists concluded the powerful antioxidant properties of UD that make it a viable chemotherapeutic treatment for breast cancer [42]. Different extraction techniques (conventional maceration extraction using ethanol and supercritical fluid extraction) were used in a comparative study to examine the antioxidant properties of UD leaves. In the ABTS experiment, it was discovered that the conventional ethanolic extract had greater antioxidant activity [43]. In a separate investigation, the antioxidant properties of several species of *Urtica* (*Urtica dioica*, *Urtica urens*, and *Urtica membranacea*) were examined using the DPPH, ABTS, and FRAP assays. In all three trials, UD had the highest levels of antioxidant activity, which supported its clinical significance relative to other *Urtica* species [44]. In another study, the antioxidant activity of a leaf extract from UD, which is high in saponins, was investigated since antioxidant activity can promote wound healing by controlling reactive oxygen species, ROS [45,46]. It was discovered that an extract from the leaves of UD has significant antioxidant activity (comparable to ascorbic acid), which was crucial for the extract’s ability to promote wound healing [46].

Colorectal cancer after stomach and esophagus cancer is the most common malignancy of the digestive system [47]. Further, since chemotherapy and/or radiotherapy in cancer patients often result in destructive adverse effects, the focus of research has been on developing effective medicinal substances that can simultaneously lessen undesirable side effects and improve the cytotoxic effect [48]. Therefore, there has been a lot of interest lately in the usage of natural products and food supplements that have anticancer characteristics [49]. Different groups have determined the cytotoxic and antitumor properties of various species of the genus *Urtica* [50,51,52].

To investigate the antitumoral activity of AEUD, we first evaluated its effect on cell proliferation in the HCT-116 colorectal cancer cell line. Cell viability was determined by the MTT assay, preliminarily, after long-term treatments (72 h). The results show that AEUD produced cytotoxic effects on HCT-116 cells with an IC_50_ value of 2.35 mg/mL.

Results of the AEUD are in accordance with that observed by Ali et al., which demonstrated that the *U. dioica* extract significantly increased the cytotoxic effects on the colorectal cancer cell line HCT-116 in comparison to the untreated control group. These effects were dose- and time-dependent, indicating that they become more pronounced with time and concentration [53]. These results were consistent with those of another investigation utilizing lymphocytes [54]. It is possible that antioxidant compounds have a linked antiproliferative impact. Through the induction of apoptosis and the cessation of the cell cycle, these compounds control the growth of tumor cells [55,56].

In numerous pathogenic conditions, the cell cycle is dysregulated. Therefore, it could be interesting to study how treatments might affect the tumor cells’ cell cycle. The flow cytometry method was used in the current investigation to identify the cell cycle phases following the treatment of cells by administration of different doses of AEUD. AEUD produced a considerable rise in the subG1 phase population, which caused apoptosis in the HCT-116 cell line, according to the flow cytometry results. The population in the S/G2/M stages was significantly increased, and the HCT-116 cancer cells’ progressive growth was inhibited. Once a cell has undergone an abnormal modification, cell cycle disruption, and uncontrollable cell division take place. Therefore, it can be concluded that treatment with *U. dioica* effectively inhibited cell growth and induced a dose-dependent apoptotic response in HCT-116 colon cancer cells.

Our data agree with a prior study, which showed that the proportion of cells in the G2/M phase increased in comparison to the control, followed by a decrease in the number of cells in the G1 and S phases [53]. The SubG0/G1 ratio increased noticeably following treatment. These findings show that the cell cycle was arrested in the G2 phase after post-*U. dioica* dichlororomethane extract treatment [53]. In a previous study of the apoptotic effect of *U.dioica,* they demonstrated that patuletin and quercetagetin suppressed proliferation with apoptotic activity in various tumor cell lines [22]. Referring to the literature, it was discovered that another substance present in *U. dioica*, m/p-hydroxybenzoic acid, which is a phenol derivative of benzoic acid and is regarded as a significant dietary phenolic agent, inhibits the proliferation of cancer cell lines by activating apoptotic pathways [23]. Furthermore, human solid-tumor cell lines from lung cancer, glioma, hepatoma, and colon cancer were prompted to undergo programmed cell death by chlorogenic acid, which has been shown to be safe for use in people [57]. Additionally, it has been observed that homovanillic acid induces death in myeloid leukemia cells both in vitro and in vivo through oxidative stress, with no deleterious effects on mice [58].

Metastasis is a significant signal of malignant cells. One crucial “hallmark” of tumor progression is cell migration and invasion [59]. Therefore, it would be advantageous to prevent invasion and migration in tumor lines [60]. Migration and invasion were assessed in our study using the wound healing assay and Transwell assay, respectively. In this investigation, we found that AEUD weakly slowed down the cell line HCT-116′s migration, as demonstrated by wound healing. In contrast, the Transwell assay revealed that our extract had a better capacity and was dose-dependent in its ability to considerably reduce the ability of HCT-116 colon cancer cells to migrate and invade. In another study, using a wound healing experiment to determine the anti-metastatic potential of *Urtica dioica* leaf extract on breast cancer cells (MCF-7, MDA-MB-231, and HFFF2), it was found that the extract healed the closure entirely in 48 h. In contrast to untreated cells, treatment of tumor cells with the IC_50_ dose of *Urtica dioica* extract reduced the rate of migration [61].

## 4. Materials and Methods

### 4.1. AEUD Preparation

Urtica dioica leaves were harvested in April 2022 in Beja, Tunisia. 3.933 kg of Urtica dioica leaves (fresh material) were collected. The following procedure was developed and used by the Tunisian group for extraction and lyophilization of samples. The leaves were dried in an oven for 48 h at a temperature of 50 °C. After drying, an amount equal to 500 g of dry material was obtained. The leaves were ground into a fine powder in a blender after 48 h of drying in an oven at 50 °C. The obtained powder was stored at room temperature in a dry, light-protected container for future use. For lyophilization, 10 g of the powder was dissolved in 100 mL of double-distilled water in a beaker and covered with aluminum foil. The covered solution was kept protected from light and agitated in a shaker for 24 h at room temperature The extracted solution was then filtered, spread as a thin layer over crystallizing dishes, covered with aluminum foil, and placed at −20 °C for 24 h. The AEUD was lyophilized for 48 h at −60 °C under a pressure of 0.01 mbar using a freeze-dryer (ModuloyD Freeze dryer, Thermo Fisher, Waltham, MA, USA). Finally, the lyophilized product (AEUD) was collected in dry, light-protected tubes for in vitro assays.

### 4.2. Non-Targeted Metabolomics Characterization of Urtica dioica Leaf Extract

#### 4.2.1. Ultra-High-Performance Liquid Chromatography

Samples were analyzed using an Easy nLC 1200 ultra-high-performance liquid chromatography system coupled to a Q-Exactive HF-X quadrupole-Orbitrap mass spectrometer (Thermo Fisher Scientific, Waltham, MA, USA). Compounds were separated on a C18 analytical column (PepMap RSLC C18, 2 µm, 100 Å, 75 µm × 50 cm; Thermo Fisher Scientific, Waltham, MA, USA), thermostatted at 40 °C. A 60-min gradient of 0.1% formic acid in acetonitrile was applied at a flow rate of 300 nL/min to achieve optimal separation.

#### 4.2.2. High-Resolution Mass Spectrometry

Data acquisition was performed in electrospray ionization (ESI) positive mode using Tune 2.9 and Xcalibur 4.1.31.9 software for instrument operation and control. External calibration was conducted using the LTQ Velos ESI Positive Ion Calibration Solution (Thermo Fisher Scientific, Waltham, MA, USA), while internal calibration utilized the polysiloxane ion signal at 445.120024 *m*/*z*. MS1 spectra were acquired at a resolution of 120,000 over a mass range of 100–1300 *m*/*z*. Data-dependent MS/MS fragmentation was performed on the five most intense precursor ions, with fragment ions generated in a higher-energy collisional dissociation (HCD) cell and detected in the Orbitrap mass analyzer at a resolution of 30,000. Dynamic exclusion of previously selected precursor ions was applied for 10 s to minimize redundancy. Automatic gain control (AGC) was employed to prevent saturation, with target values set at 3 × 10⁶ ions for MS1 and 1 × 10^5^ ions for MS2.

#### 4.2.3. Raw Data Processing

Ionization data were processed using Compound Discoverer 3.3 software (Thermo Fisher Scientific, MA, USA). The software enabled retention time alignment, detection of unknown compounds, and prediction of elemental compositions while filtering out chemical backgrounds using blank samples. Compound annotations were validated with a mass error threshold of 5 ppm. Fragmentation-based compound predictions were performed using mzCloud, ChemSpider, and a local database containing 68 compounds, as detailed in the Appendix A.

### 4.3. In Vitro Evaluation of Antibacterial Activity

#### 4.3.1. Bacterial Strains

The potential antibacterial activity of AEUD was tested by using the disc-diffusion and microdilution methods, according to guidelines recommended by the “Comité de l’Antibiogramme de la Société Française de Microbiologie” (CA-SFM), against nine pathogenic microbial strains of both Gram-negative bacteria (*Escherichia coli* ATCC 25983, *Salmonella typhimurium* ATCC14028, *Salmonella enteritidis* ATCC13076, *Pseudomonas aeruginosa* ATCC27853 and *Shigella flexneri* (clinical isolate) and Gram-positive bacteria (*Staphylococcus aureus* ATCC29213, *Enterococcus faecalis* ATCC29212 and *Bacillus cereus* ATCC 11778 and *Listeria monocytogenes* ATCC7644).

#### 4.3.2. Disc-Diffusion Test

A fresh bacterial suspension containing 10^8^ colony-forming units (CFU)/mL was spread on Muller-Hinton agar (MHA). Sterile discs (6 mm in diameter) were impregnated each with 20 μL of the tested extract dissolved in DMSO (50 mg/mL), and placed on the inoculated agar plates. Negative control prepared using DMSO was used in negative controls and Gentamicin (10 µg/disc) in positive controls. The inoculated plates were incubated at 37 °C for 24 h. Antimicrobial activity was assessed by measuring the diameter of the growth-inhibition zone in mm All tests were carried out for three sample replications, and values are the means of three replicates.

#### 4.3.3. Determination of Minimum Inhibitory Concentration (MIC)

MIC values were determined, based on a micro-well dilution method. The bacterial suspension was adjusted with sterile saline to a concentration of 10^8^ CFU/mL. The compound was dissolved in broth LB medium (100 µL) with bacterial inoculums to achieve the desired concentrations (50–0.097 mg/mL). After overnight incubation at 37 °C, the MIC value was the lowest extract concentration preventing the development of visible bacterial growth.

#### 4.3.4. Determination of Minimum Bactericidal Concentration (MBC)

After MIC identification, a portion of liquid (5 μL) from each plate well without visible growth in MIC assay, was cultured on Mueller-Hinton Agar plates and incubated at 37 °C overnight. MBC was that of the highest dilution yielding no colony formation on MHA.

### 4.4. In Vitro Evaluation of Antioxidant Activity

#### 4.4.1. Free Radical Scavenging Activity (DPPH Assay)

The DPPH free radical-scavenging activity of AEUD at different concentrations was assessed as described previously [62]. Briefly, 500 μL of each concentration of AEUD was mixed with 375 μL of 100% ethanol and 125 μL of 0.02 mM DPPH in 100% ethanol. After 60 min in the dark at room temperature, DPPH radical reduction was measured at 517 nm using a UV-visible spectrophotometer. DPPH radical-scavenging activity was calculated as follows:DPPH free radical-scavenging activity (%) = ((Ab + Ac − Ah)/Ac) × 100,
where Ab is the absorbance of the blank, Ac is the absorbance of the control reaction and Ah is the absorbance of the sample. Trolox (6-hydroxy-2,5,7,8-tetramethylchroman-2-carboxylic acid) was used as a standard. In controls, distilled water was used instead of the sample. The test was carried out in triplicate.

#### 4.4.2. Total Radical Scavenging Capacity (ABTS Assay)

The free radical-scavenging activity was determined by ABTS radical cation decolorization assay [63], ABTS solution was prepared by mixing the radical ABTS (7 mM) with aqueous potassium persulfate (2.45 mM). This mixture was kept in the dark at room temperature for 16 h before use. The working solution was prepared by then, it was filtered and diluted with ethanol to obtain a working solution with an absorbance of 0.7 ± 0.02 at 734 nm. Samples of 1mL of AEUD sat the different tested dilutions (1–100 μg/mL) and the standard was added to 3 mL of ABTS working solution. After 60 min at room temperature in the dark, absorbance was measured at 734 nm. Data are expressed as % Inhibition according to the following expression:I (%) = ((Abs blank − Abs sample)/Abs blank) × 100,
where: Abs blank = is the absorbance of the control reaction at t = 0 (containing all reagents except the test compound), and Abs sample: is the absorbance of the test compound. All determinations were performed in triplicate. Ascorbic acid was used as a reference.

#### 4.4.3. Free Reducing Antioxidant Power (FRAP)

The ability of AEUD to reduce iron was determined according to the method of Yildirim et al. [64], with slight modifications. 0.5 mL of different concentrations were mixed with 1.25 mL of potassium phosphate buffer (0.2 M, pH 6.6) and 1.25 mL of 1% (*w*/*v*) potassium ferricyanide solution. The mixtures were incubated at 50 °C for 30 min. After incubation, 1.25 mL of 10% trichloroacetic acid (*w*/*v*) was added. After centrifugation at 3000× *g* for 10 min, 1.25 mL of supernatant was mixed with 1.25 mL of 0.1% ferric chloride (*m*/*v*). After 10 min, absorbance was measured at 700 nm. BHT (butylhydroxytoluene) was used as a standard. The test was carried out in triplicate.

#### 4.4.4. Ferrous Ion-Chelating Ability Assay

The ferrous ion-chelating ability was determined according to the method of Decker and Welch [65]. Sample solutions (100 µL) were mixed with 2 mM FeCl_2_ (50 µL) and distilled water (450 µL). The reaction was initiated by adding 5 mM ferrozine (200 µL). After 10 min at room temperature, the absorbance of the Fe^2+^-ferrozine complex (with red to purple color) was measured at 562 nm. The chelating activity of the antioxidant for Fe^2+^ was calculated according to the following formula:Ferrous ion-chelating ability (%) = [(Ab + Ac) − Ah/A_0_ × 100]
where A_0_ is the absorbance of the blank; Ac is the absorbance of the sample; and Ah is the absorbance in the presence of the sample. BHT (butylhydroxytoluene) was used as a standard. The control was conducted in the same manner, except that distilled water was used instead of the sample. The test was carried out in triplicate.

### 4.5. In Vitro Evaluation of Antitumor Activity

#### 4.5.1. Cell Line and Cell Culture Conditions

In the present work, we used HCT-116 cells, a human epithelial colorectal carcinoma, purchased from the American Type Culture Collection (ATCC). Cells were cultured in a humidified incubator with an atmosphere of 5% CO_2_ at 37 °C and cultivated in DMEM containing glucose (4.5 g/L) supplemented with 10% fetal bovine serum (FBS) in the presence of penicillin and streptomycin.

#### 4.5.2. Cell Viability Assay

MTT (3-[4,5-dimethyl-2-thiazolyl]-2,5-diphenyl tetrazolium bromide), a yellow, water-soluble chemical, was used to test the toxicity of AEUD on HCT-116 cells on quadruplicates. MTT was reduced by mitochondrial reductase of live cells to blue formazan salt and absorbance at 550 nm was measured with a plate reader. In brief, cells were seeded into 96-well plates at 2 × 10^3^ cells per well. Serial dilutions of the AEUD concentrations were added to the wells in a complete medium and incubated at 37 °C and 5% CO_2_ for 72 h. The medium was removed, and a new medium was added to all wells to prevent any color interference. Then, MTT stock solution (5 mg/mL) was added to each well. After 4 h of incubation at 37 °C, 150 µL of acidified isopropanol was added to each well and absorbance was measured using an Eon Microplate Spectrophotometer from Bio-Tek Instruments (Winooski, VT, USA). Data were collected by Gen5 software from the same manufacturers at a wavelength of 550 nm. The IC_50_ values (concentration resulting in 50% of cell growth inhibition) were estimated from the growth curves.

#### 4.5.3. Cell Cycle Analysis

Using the propidium iodide staining procedure, the cell cycle was examined through the changes in DNA content during cell cycle progression, as determined with flow cytometry. HCT-116 cells were seeded at the density of 1.5 × 10^5^ cells/well, with 1.5 mL culture medium in 6-well plates. After 24h, cells were treated with various doses of AEUD (4, 2, and 1 mg/mL), using 10 µM 2-methoxystradiol as positive control. The cells were removed after 24 h of incubation, washed twice with PBS, and then fixed in ice-cold 70% ethanol at 4 °C for one h. Then, 5 μL of RNase (10 mg/mL) and 20 μL of propidium iodide (1 mg/mL) were added to cell suspensions. After 30 min of incubation in the dark, the obtained cell suspensions were filtered through a 0.4 µm cell strainer to remove any cell aggregates and cell cycle analysis was carried out using a flow cytometer (BD FACSVerse, BD, Franklin Lakes, NJ, USA). The percentages of cells present in the sub G1, G0/G1, S/G2/M, and G1/G2 phases were calculated using the Kaluza software (2.3 release).

#### 4.5.4. Migration/Mobility (“Wound Healing”) Assay

A relatively basic experiment, the wound healing assay, can be used to determine the ability of cells to migrate under various conditions. In this experiment, a portion of the plate’s cells are mechanically removed, creating an opening for additional cells to move toward. The incubation times for this experiment were 4, 7, 24, and 48 h. Mitomycin B (10 ng/mL) is required for lengthier incubations when cell proliferation could occur to prevent DNA synthesis and cell growth.

A transverse scratch in the monolayer was made using a sterile 200 µL pipette tip after the HCT-116 colon cancer cells were seeded in 6-well plates and cultured at 37 °C until they reached 90% confluence. The cells were treated with 1, 2, and 4 mg/mL of AEUD after being washed with PBS to eliminate non-adherent cells. The cell cultures were photographed using a Nikon DS-Ri2 attached to a Nikon Eclipse Ti microscope (Nikon, Tokyo, Japan) at 0, 4, 7, 24, and 48 h, and migration rate % was calculated. Image J software was used to measure the cell-free regions.

#### 4.5.5. Transwell Invasion Assay

Matrigel-coated transwells were used to determine the invasive potential of ECs. HCT-116 cancer colon cells were grown until 70–80% of confluency and then incubated overnight in FBS-free media supplemented with 0.1% BSA. Next, cells were pre-treated with DMSO or different doses of AEUD for 24 h, at concentrations 1, 2, and 4 mg/mL. In the lower compartment, the medium was placed with FBS, which acted as a chemoattractant, except for the negative control. Then, the media on the top of the transwells were removed by pipetting and using cotton swabs. Adhered migrating and invading cells in the lower side of the transwells were washed with PBS, fixed in clean wells containing 300 μL paraformaldehyde, and stained with 0.1% crystal violet, for 20 min. Finally, the number of invading cells was counted. For this, photos of five randomly selected representative fields were taken using a Nikon DS-Ri2 connected to a Nikon Eclipse Ti microscope (Nikon, Tokyo, Japan). The number of invading cells was counted in each photograph and the mean was calculated for each condition. Cell number in control condition with FBS in the bottom well was considered as 100%.

### 4.6. Numerical Data Analysis

The GraphPad Prism 8 software was used to analyze the data. The data are shown as mean SD. One-way analysis of variance (ANOVA) was used to evaluate the variances for all data. At *p* < 0.05, differences were deemed statistically significant.

## 5. Conclusions

A qualitative biochemical composition of AEUD is provided. Our in vitro results demonstrated the antibacterial and antioxidant properties of *Urtica dioica* leaf aqueous extract. Additionally, the AEUD prevented HCT-116 colon cancer cells from proliferating and causing cell cycle arrest at the G2 phase. Treatment with AEUD reduced the migration and invasion of HCT-116 cells, suggesting AEUD’s capacity to prevent colon cancer metastasis. This might be due to the plant’s phenol content, which seems mostly responsible for the many biological effects seen in many *Urtica* species. It should be warranted to perform additional experiments to assess this plant’s potential in vivo anticancer effects.

## Figures and Tables

**Figure 1 ijms-26-01220-f001:**
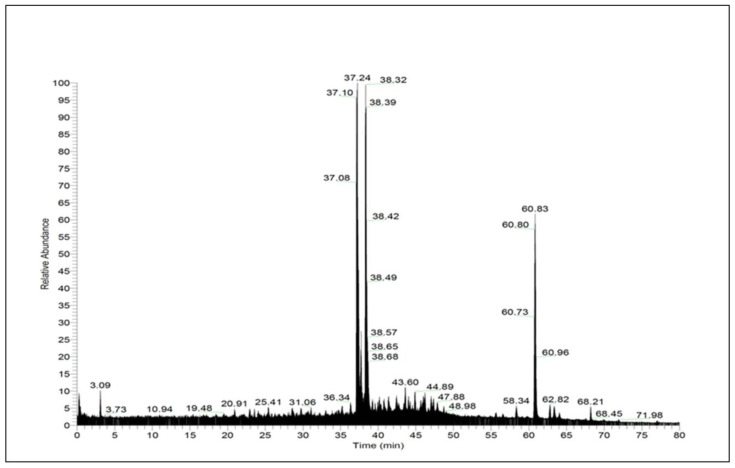
Total ion chromatogram for *Urtica dioica* aqueous leaf extract acquired in full scan data-dependent acquisition MS2.

**Figure 2 ijms-26-01220-f002:**
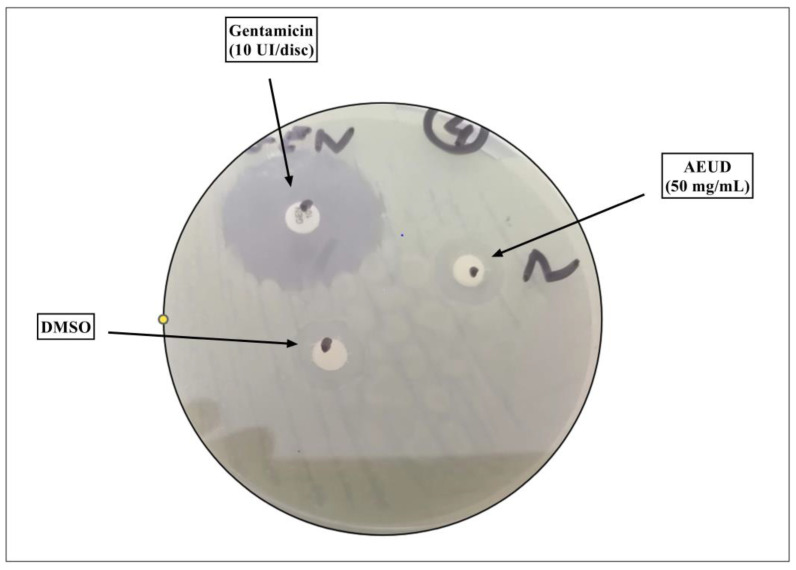
Inhibitory effect of AEUD and Gentamicin on the reference strain *Enterococcus faecalis* ATCC 29212.

**Figure 3 ijms-26-01220-f003:**
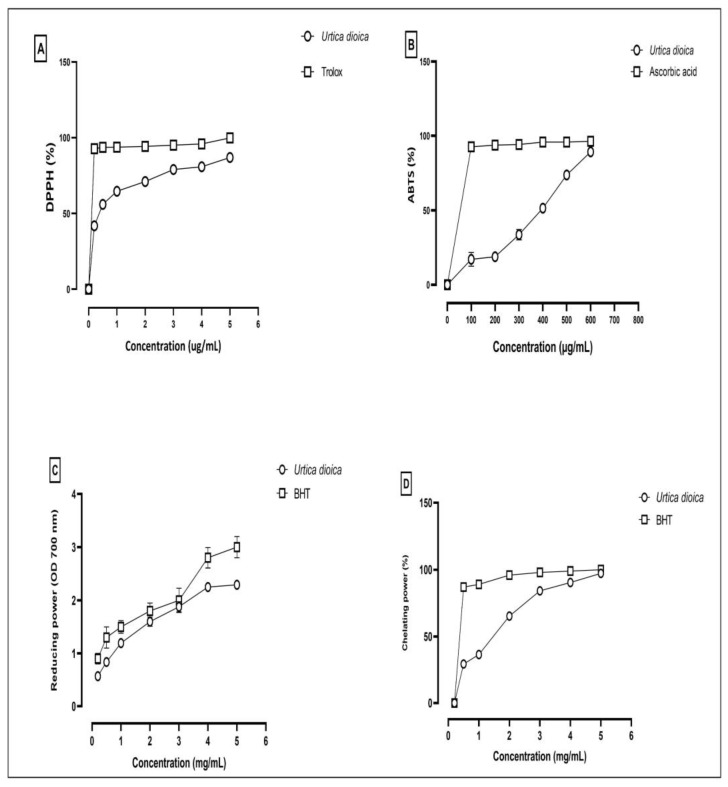
Antioxidant activity of aqueous extract of *U. dioica* leaves. (**A**) Free radical scavenging activity (DPPH assay). (**B**) Total radical scavenging capacity ABTS assay. (**C**) Ferric reducing antioxidant power (FRAP). (**D**) Ferrous ion-chelating ability assay.

**Figure 4 ijms-26-01220-f004:**
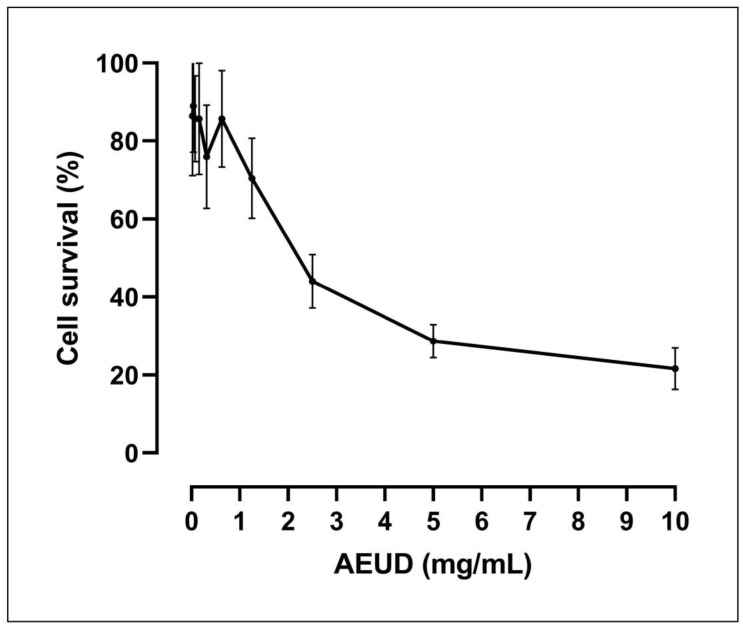
Survival curves for HCT-116 tumor cells cultured for three days in the presence of different doses of AEUD. Data are means ± S.D. of three different, independent experiments with 4 replicates each.

**Figure 5 ijms-26-01220-f005:**
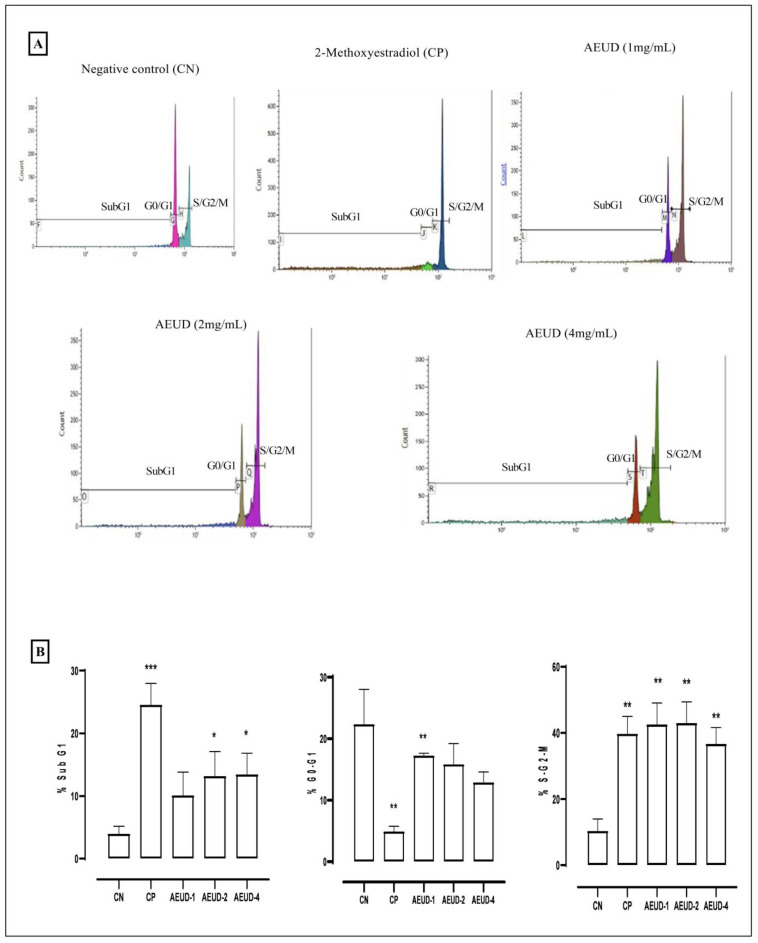
Effects of AEUD on the phases of cell cycle distribution in colon cancer cells HCT-116. Cells are distributed into the different phases of the cell cycle according to their DNA content: Sub G1 cells are <2n, G0/G1 cells are 2n, S cells are >2n but <4n, and G2/M-phase cells are 4n. (**A**) Representative flow cytometric histograms of cell cycle phases (Sub-G1, G0/G1, and S/G2/M). (**B**) Relative percentage population. CN (negative control); CP (positive control: methoxyestradiol); AEUD-1 (aqueous extract of *Urtica dioica* 1 mg/mL); AEUD-2 (aqueous extract of *Urtica dioica* 2 mg/mL); AEUD-4 (aqueous extract of *Urtica dioica* 4mg/mL). Data were acquired from three independent biological replicates and represented as mean ± SD. Statistical significance as compared with negative control values, according to a two-sided unpaired Student’s *t* test: * *p* < 0.05, ** *p* < 0.005, *** *p* < 0.0005.

**Figure 6 ijms-26-01220-f006:**
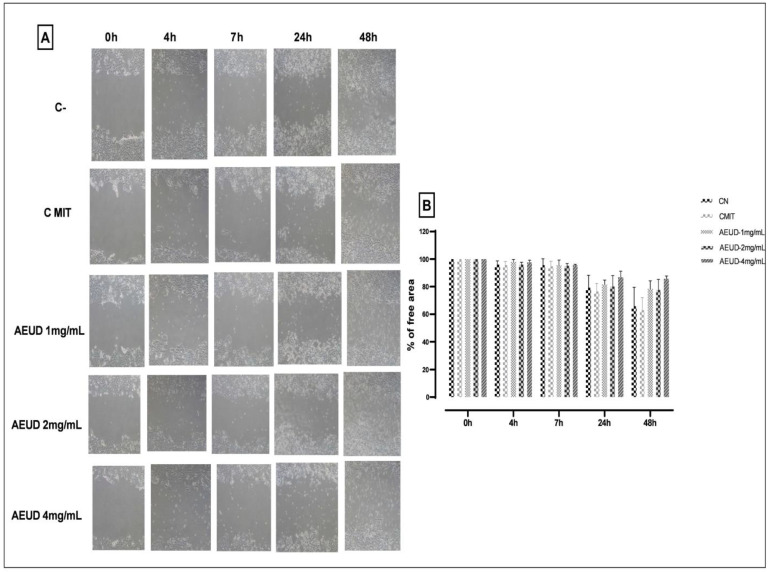
Wound healing assay (**A**) Representative images of the effect of AEUD on the migratory potential of HCT-116 colorectal carcinoma cells, (**B**) Relative quantification (as a percentage) of the cell-free areas of samples in the presence of the AEUD in different doses. Negative control (C-), negative control with mitomycin (CMIT), and cells in the presence of different doses of AEUD (1,2 and 4mg/mL) and mitomycin at times 0, 4, 7, 24, and 48 h. Data were acquired from three independent biological replicates and represented as mean ± SD.

**Figure 7 ijms-26-01220-f007:**
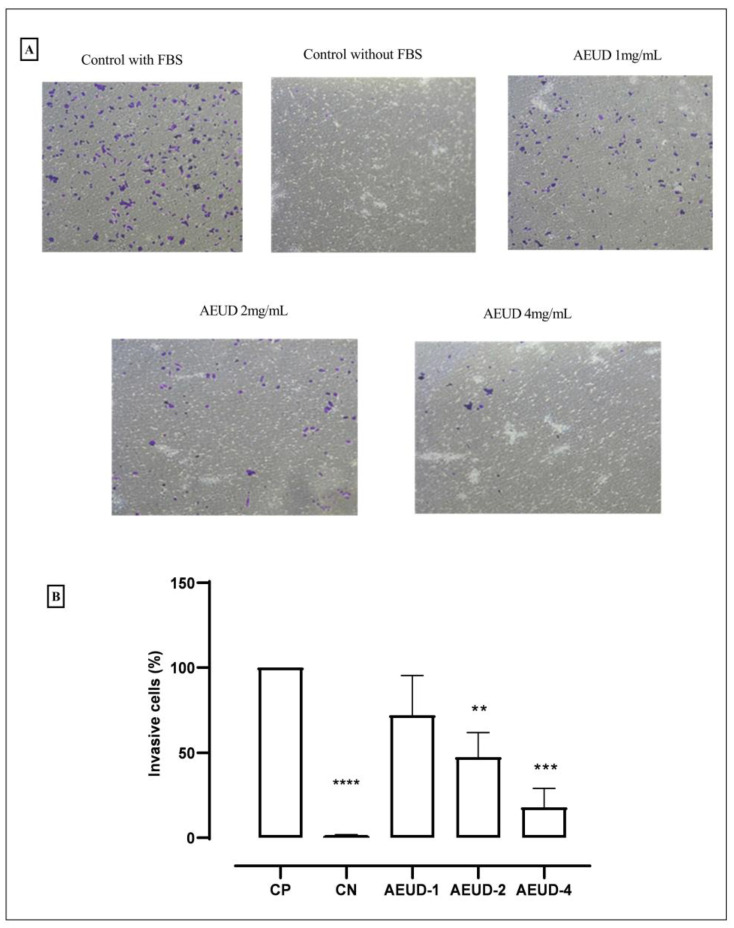
Invasion transwell assay (**A**) Representative images of the effect of AEUD on invasion potential of HCT-116 colorectal carcinoma cells. (**B**) Relative quantification of invasive ability (%) of HCT-116 by transwell assay analysis: (CP: control with FBS; CN: control without FBS). Data were acquired from three independent biological replicates and represented as mean ± SD. Statistical significance as compared with negative control values, according to a two-sided unpaired Student’s *t* test: ** *p* < 0.05, *** *p* < 0.005, **** *p* < 0.0005.

**Table 1 ijms-26-01220-t001:** List of identified compounds from *Urtica dioica* aqueous leaf extract *.

Compound	Molecular Formula	Annotation MW	*m*/*z*	Calculated MW	Mass Error (ppm)	RT (min)	Reference Ion
2-Hydroxycinnamic acid	C9 H8 O3	164.04734	165.05466	164.04738	0.21	46.229	[M+H]+1
3,4-Dihydroxybenzaldehyde	C7 H6 O3	138.03169	139.03893	138.03165	−0.33	45.023	[M+H]+1
4-Hydroxycoumarine	C9 H6 O3	162.03169	163.03888	162.03160	−0.56	61.499	[M+H]+1
4-Methylumbelliferone hydrate	C10 H8 O3	176.04734	209.08083	176.04728	−0.38	25.641	[M+H+MeOH]+1
alpha-Bisabolol	C15 H26 O	222.19837	223.20572	222.19844	0.35	47.887	[M+H]+1
alpha-Bisabolol, acetate	C17 H28 O2	264.20893	247.20554	264.20872	−0.80	38.134	[M+H-H_2_O]+1
alpha-Farnesene	C15 H24	204.18780	205.19513	204.18786	0.27	46.802	[M+H]+1
alpha-Pinene	C10 H16	136.12520	137.13272	136.12544	1.79	0.094	[M+H]+1
Angelic acid	C5 H8 O2	100.05243	101.06010	100.05283	3.98	41.247	[M+H]+1
Azulene	C10 H8	128.06260	129.07001	128.06273	1.02	32.950	[M+H]+1
Bisabolol oxide A	C15 H26 O2	238.19328	239.20044	238.19316	−0.49	35.740	[M+H]+1
Caffeic acid	C9 H8 O4	180.04226	181.04964	180.04250	1.34	46.627	[M+H]+1
Catechol	C6 H6 O2	110.03678	111.04445	110.03717	3.57	38.796	[M+H]+1
Chamazulene	C14 H16	184.12520	185.13251	184.12523	0.16	42.687	[M+H]+1
Ethyl protocatechuate	C9 H10 O4	182.05791	183.06531	182.05803	0.67	14.681	[M+H]+1
p-Coumaric acid	C9 H8 O3	164.04734	165.05460	164.04732	−0.15	20.856	[M+H]+1

* Molecular formula: Assigned elemental composition. Annotation MW: Theoretical molecular weight of assigned annotation. *m*/*z*: *m*/*z* value of the leftmost isotopic peak of the most common adduct ion for this compound. Calculated MW: Neutral mass in Da retrieved from the measured leftmost isotopes of related compounds. Mass error (ppm): Difference between measured and theoretical molecular weight of assigned annotation in ppm. RT (min): Retention time in min. Reference ion: The most common adduct ion for this compound.

**Table 2 ijms-26-01220-t002:** In vitro antibacterial activity of aqueous extract of *U. dioica* by the diffusion disk method.

Strains	Zone of Inhibition (mm) ^1^	
	AEUD (20 µL/disc)	Gentamicin (10 UI/disc)
**Gram positive**		
*S. aureus* ATCC29213	6 ± 0	30
*E. faecalis* ATCC29212	10 ± 1	33
*B. cereus* ATCC11778	6 ± 0	30
*L. monocytogenes* ATCC7644	6 ± 0	30
**Gram negative**		
*E. coli* ATCC25983	6 ± 0	30
*P. aeruginosa* ATCC27853	13 ± 1	30
*S. enteritidis* ATCC13076	6 ± 0	31
*S. typhimirium* ATCC14028	14 ± 0	37
*Shigella flexneri*	6 ± 0	33

^1^ Values represent means ± standard deviations for triplicate experiments.

**Table 3 ijms-26-01220-t003:** Minimum inhibitory concentration and minimum bactericidal concentration of aqueous extract of *U. dioica* against tested strains.

Strains	MIC (mg/mL)	MBC (mg/mL)	Bacterial Activity
**Gram positive**			
*S. aureus* ATCC29213	6.25	50	Bacteriostatic
*E. faecalis* ATCC29212	0.39	0.39	Bactericidal
*B. cereus* ATCC 11778	0.39	0.39	Bactericidal
*L. monocytogenes* ATCC7644	1.56	1.56	Bactericidal
**Gram negative**			
*E. coli* ATCC25983	0.195	50	Bacteriostatic
*P. aeruginosa* ATCC27853	0.39	50	Bacteriostatic
*S. enteritidis* ATCC13076	12.5	25	Bactericidal
*S. typhimirium* ATCC14028	1.562	1.562	Bactericidal
*Shigella flexneri*	12.5	25	Bactericidal

## Data Availability

Data will be made available on request.

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
