# Peer review of "Urtica dioica Aqueous Leaf Extract: Chemical Composition and In Vitro Evaluation of Biological Activities"

_ijms, 2025, doi:10.3390/ijms26031220_

Round 1

Reviewer 1 Report

Comments and Suggestions for Authors

This manuscript has made significant contributions to the study of the chemical constituents and biological activities of aqueous extracts from Urtica leaves. Firstly, the research methodology is scientific and rigorous, utilizing advanced ultra-performance liquid chromatography-mass spectrometry (UPLC-MS) technology to analyze the chemical components. A range of biologically active compounds are successfully identified, providing a solid foundation for further research into their biological activities. Secondly, the evaluation of biological activities is comprehensive and systematic, addressing various aspects such as antibacterial, antioxidant, and anti-tumor effects. The experimental design is sound, and the results are reliable, offering substantial scientific value. Furthermore, the discussion section delves deeply into the experimental results and explores potential mechanisms in light of existing literature. The logical flow is clear, and the arguments are well-supported, enhancing readers' understanding of the research's significance.

Although the paper excels in many areas, there are opportunities for further enhancement. Firstly, in the chemical composition analysis, it would be beneficial to incorporate additional reference materials or standards. This could improve the accuracy and comparability of both qualitative and quantitative results, thereby ensuring the reliability of the research data. Secondly, in the anti-tumor activity study, it is recommended to conduct further molecular biology experiments, such as assessing changes in gene expression and signaling pathways. This would help to elucidate the underlying mechanisms of action and provide a more detailed theoretical foundation for future drug development and clinical applications.

Author Response

REPLY TO REVIEWER 1 (Reviewer’s comments in black. Our reply in blue)

This manuscript has made significant contributions to the study of the chemical constituents and biological activities of aqueous extracts from Urtica leaves. Firstly, the research methodology is scientific and rigorous, utilizing advanced ultra-performance liquid chromatography-mass spectrometry (UPLC-MS) technology to analyze the chemical components. A range of biologically active compounds are successfully identified, providing a solid foundation for further research into their biological activities. Secondly, the evaluation of biological activities is comprehensive and systematic, addressing various aspects such as antibacterial, antioxidant, and anti-tumor effects. The experimental design is sound, and the results are reliable, offering substantial scientific value. Furthermore, the discussion section delves deeply into the experimental results and explores potential mechanisms in light of existing literature. The logical flow is clear, and the arguments are well-supported, enhancing readers' understanding of the research's significance.

Although the paper excels in many areas, there are opportunities for further enhancement. Firstly, in the chemical composition analysis, it would be beneficial to incorporate additional reference materials or standards. This could improve the accuracy and comparability of both qualitative and quantitative results, thereby ensuring the reliability of the research data. Secondly, in the anti-tumor activity study, it is recommended to conduct further molecular biology experiments, such as assessing changes in gene expression and signaling pathways. This would help to elucidate the underlying mechanisms of action and provide a more detailed theoretical foundation for future drug development and clinical applications.

First of all, thank you for your kind comments, suggestions, and criticism. This study employed a non-targeted approach, which is specifically designed for the discovery and characterization of unknown compounds in complex samples. In this type of analysis, the use of reference standards is not standard practice, as the primary goal is not the accurate quantification of specific compounds but rather a broad, exploratory assessment to identify patterns. Compound annotations were validated using well-established tools, such as mzCloud, ChemSpider and local libraries, in line with best practices in the field. This approach was exploratory by design, with the goal of identifying candidates that could be used in future targeted studies, where the inclusion of reference standards would in fact be more relevant. To address any potential misunderstandings, we have added a brief explanation of this methodology in Introduction and Materials and Methods sections to provide further clarity to readers.

Reviewer 2 Report

Comments and Suggestions for Authors

Dear authors, 

thank you for presenting the possibilities of urtica dioica use. 

Overal opinion

the peasant - rural common knowledge which uses the urtica as natural remedy for medical purposes is now presented in the scientific way. 

the paper is well written, using the appropriate research design and presenting the results are scientifically sound and  that will be usefull for scientists. 

Particular changes to be made: 

in the intrduction some references should be added, especcially about the procedures of urtica leaves collection, and processing, how did other authors and you did it ?

the methods should be put before results

now results are nr. 2 and materials nr 4. 

in materials lines 365 onwards to 372

How much urtica was collected (in mg, g, Kg?)

what was the initial and final weight and volumes

How much of the leaves were grounded (mg, g, Kg?)

what was the initial and final weight and volumes?

what was the ration of usefull leaves from the amount of plants (20 % or more )

how much from fresh material responded to dried material ?

how much double destilled water was used per mg/g of urtica dry mater

(e.g powder mixture) show the initial and final weight and volumes

How was the extracted solution concentrated (how much material, how long, what pressure, temperature, duration)

what was the initial and final weight and volumes ?

Was this your own method of AUED preparation ? then describe in particular all above terms, 

if not describe them all from adequate references. 

how did you and how long bathed the extract (temperature, time.. )

how did you lyophilized the extract (time, weight) 

what was the initial and final weight and volumes

Overall if the method was developed during the study add values obtained from your method and validate your method, e.g. with accurancy and precision tests. 

Results

authors may widen the figures 5 and 6 so the results are more visible

(if it is possible)

kind regards, 

the reviewer

Author Response

REPLY TO REVIEWER 2 (Reviewer’s comments in black. Our reply in blue)

Dear authors, 

thank you for presenting the possibilities of urtica dioica use. 

Overal opinion

the peasant - rural common knowledge which uses the urtica as natural remedy for medical purposes is now presented in the scientific way. 

the paper is well written, using the appropriate research design and presenting the results are scientifically sound and  that will be usefull for scientists. 

Thank you for your kind comments, and positive evaluation of our manuscript.

Particular changes to be made: 

in the introduction some references should be added, especcially about the procedures of urtica leaves collection, and processing, how did other authors and you did it ?

the methods should be put before results

now results are nr. 2 and materials nr 4. 

Thank you, but this is not the case. We have used the template provided by the journal and in this template (and in articles already published) the Methods section is to be included after Results and Discussion sections.

in materials lines 365 onwards to 372

How much urtica was collected (in mg, g, Kg?)

what was the initial and final weight and volumes

How much of the leaves were grounded (mg, g, Kg?)

what was the initial and final weight and volumes?

what was the ration of usefull leaves from the amount of plants (20 % or more )

how much from fresh material responded to dried material ?

how much double destilled water was used per mg/g of urtica dry mater

(e.g powder mixture) show the initial and final weight and volumes

How was the extracted solution concentrated (how much material, how long, what pressure, temperature, duration)

what was the initial and final weight and volumes ?

Was this your own method of AUED preparation ? then describe in particular all above terms, 

if not describe them all from adequate references. 

how did you and how long bathed the extract (temperature, time.. )

how did you lyophilized the extract (time, weight) 

what was the initial and final weight and volumes

We have added now all the relevant information for you requested in section 4.1 

Overall if the method was developed during the study add values obtained from your method and validate your method, e.g. with accurancy and precision tests

We would like to clarify that this study utilized a non-targeted metabolomics approach, which differs from targeted methods in terms of validation and applicability of metrics like accuracy and precision. The method used was not developed de novo in this study but is based on well-established and widely accepted protocols in the field of non-targeted metabolomics. 

In non-targeted metabolomics, the primary objective is to provide a comprehensive and unbiased metabolic profile, rather than quantifying specific compounds with high precision. Nevertheless, we ensured the robustness of the methodology through several quality control measures.  Mass accuracy was maintained using external and internal calibrations as detailed in the Materials and Methods section, remaining below 5 ppm for all identified compounds as shown in Supplementary Material. Blank and quality control samples were analyzed throughout the process to monitor instrument performance and ensure reproducibility. Additionally, compound annotations were validated through spectral matching using established tools such as mzCloud and ChemSpider. 

Results

authors may widen the figures 5 and 6 so the results are more visible

(if it is possible)

Done.

Reviewer 3 Report

Comments and Suggestions for Authors

Dear authors, 

The article is very interesting and relevant. However, there was a question that should be clarified

1. In the section "Results" authors discusses about tumor cells and the healing process studied with them...

2.The research methods section "4.5. In vitro evaluation of antitumor activity" -  has the following paragraphs : Cell line and cell culture condition;  Cell viability assay, Cell cycle assay, Wound healing assy, Transwell  Invasion Assay". The wound healing assay was performed with the same tumor cells culture? If, so, is this a good solution because the cells are tumor-like and cannot show a true healing process. Or was another cell culture used to assess healing?

Were the cells with which the effect of the plant on healing was studied not named in the methods? Because these should be cultures of fibroblasts or normal but not tumor cells used for evaluation for healing process. 

Author Response

REPLY TO REVIEWER 3 (Reviewer’s comments in black. Our reply in blue)

The article is very interesting and relevant. However, there was a question that should be clarified

  1. In the section "Results" authors discusses about tumor cells and the healing process studied with them...

2.The research methods section "4.5. In vitro evaluation of antitumor activity" -  has the following paragraphs: Cell line and cell culture condition;  Cell viability assay, Cell cycle assay, Wound healing assy, Transwell  Invasion Assay". The wound healing assay was performed with the same tumor cells culture? If, so, is this a good solution because the cells are tumor-like and cannot show a true healing process. Or was another cell culture used to assess healing?

Were the cells with which the effect of the plant on healing was studied not named in the methods? Because these should be cultures of fibroblasts or normal but not tumor cells used for evaluation for healing process. 

First of all, thank you for your kind comments, suggestions, and positive criticism. Regarding your points 1 and 2, please take into account that the “wound healing” assay is not a real healing assay but a very used assay to test the effects of compounds on the migration/mobility of cells. To avoid misunderstanding, we have changed the title of section 4.5.4 to “Migration/mobility (“wound healing”) Assay”. We have also changed the mention to this assay in the section 2.6. The cell used are the same colon cancer cells used for the rest of assays with cell cultures.